# The Effects of Omeprazole on the Neuron-like Spiking of the Electrical Potential of Proteinoid Microspheres

**DOI:** 10.3390/molecules29194700

**Published:** 2024-10-04

**Authors:** Panagiotis Mougkogiannis, Andrew Adamatzky

**Affiliations:** Unconventional Computing Laboratory, University of the West of England, Bristol BS16 1QY, UK; andrew.adamatzky@uwe.ac.uk

**Keywords:** neuromorphic computing, omeprazole, proteinoids, Izhikevich neuron model, bio-inspired computing, neuromodulation, pharmacological computing, adaptive systems, neuron firing patterns

## Abstract

This study examines a new approach to hybrid neuromorphic devices by studying the impact of omeprazole–proteinoid complexes on Izhikevich neuron models. We investigate the influence of these metabolic structures on five specific patterns of neuronal firing: accommodation, chattering, triggered spiking, phasic spiking, and tonic spiking. By combining omeprazole, a proton pump inhibitor, with proteinoids, we create a unique substrate that interfaces with neuromorphic models. The Izhikevich neuron model is used because it is computationally efficient and can accurately simulate the various behaviours of cortical neurons. The results of our simulations show that omeprazole–proteinoid complexes have the ability to affect neuronal dynamics in different ways. This suggests that they could be used as adjustable components in bio-inspired computer systems. We noticed a notable alteration in the frequency of spikes, patterns of bursts, and rates of adaptation, especially in chattering and triggered spiking behaviours. The findings indicate that omeprazole–proteinoid complexes have the potential to serve as adaptable elements in neuromorphic systems, presenting novel opportunities for information processing and computation that have origins in neurobiological principles. This study makes a valuable contribution to the expanding field of biochemical neuromorphic devices and establishes a basis for the development of hybrid bio-synthetic computational systems.

## 1. Introduction

Unconventional computing is a field that investigates different ways of processing information and performing computations, going beyond the use of classic silicon-based technologies [1]. Bio-inspired computing systems have attracted considerable attention in this field because of their promising potential for the energy economy, versatility, and parallel processing capabilities [2]. Recent progress in the field has resulted in the examination of several biological and chemical materials for use in computation. These include DNA-based computing [3], reaction–diffusion systems [4], and neuromorphic computing [5]. The latter, which draws inspiration from the form and function of biological neural networks, has demonstrated significant potential in tasks such as pattern recognition and adaptive learning [6]. Our research specifically examines the relationship between omeprazole–proteinoid complexes and Izhikevich neuron models.

Omeprazole, primarily recognised as a proton pump inhibitor (PPI) for reducing gastric acid secretion in various gastrointestinal disorders [7], has recently attracted attention for its potential effects on neuronal function. Numerous studies indicate that PPIs, such as omeprazole, might have neuroprotective properties and could affect neuronal activity in addition to their primary gastric functions [8]. Research indicates that omeprazole is capable of crossing the blood–brain barrier and interacting with multiple ion channels and receptors within the central nervous system [9]. Additionally, PPIs have been shown to influence intracellular pH and membrane potential across different cell types, including neurons [10]. The findings have prompted investigations into the potential applications of omeprazole in neurological conditions and its effects on neuronal-like systems [11]. Proteinoid microspheres, as simplified models for protocells, demonstrate electrical potential changes akin to neuronal spiking [12]. The application of omeprazole enables an investigation into the impact of a clinically relevant compound, recognised for its effects on proton pumps, on these primitive, neuron-like behaviours. This approach improves our understanding of the physiological effects of omeprazole and offers insights into the essential characteristics of protocellular systems and their reactions to pharmaceutical agents. The study of omeprazole’s impact on proteinoid microspheres holds significance for unconventional computing. Unconventional computing paradigms, including those using biological or chemical systems, aim to take advantage of the computational capabilities of unconventional substrates [13]. Proteinoid microspheres demonstrate neuron-like spiking behaviour, indicating their potential as foundational components for bio-inspired computing systems [14]. Investigating the modulation of the electrical properties of microspheres by omeprazole provides insights into controlling and programming unconventional computing elements [15]. This research could advance the development of innovative computational architectures that use the distinct characteristics of protocellular systems, possibly resulting in more efficient or specialised computing solutions for specific problem types [16].

Omeprazole is combined with proteinoids, which are thermal proteins capable of forming microspheres and have been extensively explored in the field of artificial cells [17]. The Izhikevich neuron model, known for its computational efficacy and capacity to replicate a diverse array of neuronal firing patterns [18], serves as our framework for exploring the computational characteristics of these biochemical complexes. We analyse five specific neuronal behaviours: accommodation, chattering, triggered spiking, phasic spiking, and tonic spiking. These behaviours indicate diverse ways in which biological neural networks perform computations [19]. Our objective is to discover new methods for information processing and computing by studying the effects of omeprazole–proteinoid complexes on neuronal dynamics. This technique connects the fields of pharmacology, proteinoid chemistry, and neuromorphic computing, potentially creating opportunities for the advancement of adaptive, bio-inspired computational systems [13,15,20]. Our research adds to the expanding field of neuromorphic substrates and could impact the development of future hybrid bio-synthetic computational architectures. Moreover, it offers valuable information about the possible neuromodulatory impacts of pharmacological drugs when paired with proteinoids, which could have significant implications for the fields of computing and neuropharmacology. To fully understand the spiking behaviour and signal-processing abilities of the omeprazole–proteinoid complex, it is essential to have a solid foundation of the molecular structures of the proteinoid (L-Glu:L-Asp:L-Phe) and omeprazole. Figure 1 depicts these structures and their possible interactions. The amino acid composition of the proteinoid offers a range of functional groups that can engage in hydrogen bonding and other non-covalent interactions with omeprazole. These interactions are highly likely to have a crucial impact on the voltage-sensitive conformational changes and charge redistribution pathways that are described in our mechanistic model. The precise configuration of atoms and chemical bonds in omeprazole, specifically its sulphoxide group and benzimidazole ring, could potentially impact the electrical characteristics of the proteinoid. This influence may arise from the modulation of local pH gradients or the modification of conductive pathways within the complex.

The Izhikevich neuron model is described by a system of two differential equations:(1)$$\frac{dv}{dt}=0.04v^{2}+5v+140-u+I$$
(2)$$\frac{du}{dt}=a\left(bv-u\right)$$
with the auxiliary after-spike resetting:(3)$$\mathrm{if}\;vs.\geq 30\;\mathrm{mV},\;\mathrm{then}\;\left\{\begin{array}{c} v\leftarrow c\\ u\leftarrow u+d \end{array}\right.$$
Here, *v* represents the membrane potential of the neuron, *u* is a recovery variable, and *I* is the input current. The parameters *a*, *b*, *c*, and *d* are dimensionless parameters that can be adjusted to produce various types of neuronal behaviour:*a*: the time scale of the recovery variable *u*;*b*: the sensitivity of the recovery variable *u* to the subthreshold fluctuations of the membrane potential *v*;*c*: the after-spike reset value of the membrane potential *v*;*d*: after-spike reset of the recovery variable *u*.

In the Izhikevich model, the 30 mV threshold signifies the spike initiation threshold, founded in empirical findings regarding typical mammalian cortical neurons [18]. When *v* reaches or exceeds this threshold, it is considered that the neuron has generated an action potential. The model thereafter executes a swift reset of the membrane potential to a lower level (*c*) and an elevation of the recovery variable (*u* is increased by *d*), emulating the refractory time seen in biological neurons [19]. For v<30 mV, the neuron’s dynamics are controlled by the continuous differential Equations (Equation 1) and (Equation 2), which characterise the subthreshold behaviour of the neuron. The subthreshold regime includes methods like synaptic input integration and subthreshold oscillations, which are essential for neural information processing [22]. This model’s adaptability enables the simulation of diverse neuronal firing patterns found in cortical neurons by the modification of parameters *a*, *b*, *c*, and *d* [18,19].

Figure 2 illustrates the conceptual framework of our unconventional computing approach, showing how omeprazole–proteinoid complexes interact with the Izhikevich neuron model to potentially modulate various neuronal behaviours.

Proton pump inhibitors (PPIs) are a type of drug that decreases the production of acid in the gut by permanently blocking the hydrogen/potassium adenosine triphosphatase enzyme system in the cells of the stomach lining [10]. Omeprazole, the subject of this investigation, is among a number of proton pump inhibitors (PPIs) presently being used in clinical practice. Table 1 displays a comparison of omeprazole and other prevalent PPIs, emphasising their chemical formulae, half-lives, and pKa values. Although these medications have a similar way of working, they have slight variations in their pharmacokinetic and pharmacodynamic characteristics [23,24,25,26]. We selected omeprazole for this work due to its extensive use and well-established characteristics, which make it an excellent candidate for investigating the potential neuromodulatory effects in conjunction with proteinoid structures.

## 2. Materials and Methods

### 2.1. Synthesis of Omeprazole–Proteinoid Complex

The omeprazole–proteinoid complex was created by combining separately prepared solutions of omeprazole and proteinoid. Three distinct omeprazole solutions were formulated by dissolving varying quantities of omeprazole (7.2 mg, 8.3 mg, and 13 mg) in 5 mL of dimethyl sulphoxide (DMSO) sourced from Sigma Aldrich (CAS: 67-68-5, EC: 200-664-3, MW: 78.13 g/mol). Omeprazole (Merck, CAS: 73590-58-6, MW: 345.42 g/mol) was precisely weighed using an analytical balance and transferred to clean and dry beakers. These mixtures were then agitated with a magnetic stirrer at room temperature until the complete dissolution of omeprazole was achieved. During the course of the study, a 5 mL proteinoid solution comprising L-glutamic acid (L-Glu), L-phenylalanine (L-Phe), and L-aspartic acid (L-Asp) was prepared in a separate beaker, ensuring thorough dissolution in the aqueous medium. The DMSO-based omeprazole solutions were then gradually introduced into the aqueous proteinoid solution. The resulting mixtures were gently stirred using a magnetic stirrer for 5–10 min to ensure the complete and uniform blending of the omeprazole–proteinoid complexes. Following this process, the synthesised omeprazole–proteinoid complexes were ready for subsequent characterisation and analysis.

### 2.2. Electrochemical Analysis of the Omeprazole–Proteinoid Complex

The experimental setup to assess the electrochemical properties of the omeprazole–proteinoid complex is shown in Figure 3. This arrangement employs two needle electrodes constructed from platinum- and iridium-coated stainless steel wires (Spes Medica S.r.l., Genova, Italy). These electrodes are immersed in the omeprazole–proteinoid solution, maintained at a fixed distance of 10 mm apart. To capture voltage responses with high precision, the electrodes are connected to a 24-bit ADC data recorder (Pico Technology), enabling the detection of minute voltage fluctuations in the microvolt range. The sample solution is contained within a vessel placed on a temperature-controlled heating block, allowing for precise thermal regulation throughout the experiment. The heating block is synchronised with the data logger to enable the simultaneous recording of thermal and electrical characteristics. Electrical stimuli mimicking Izhikevich neurons are applied to the omeprazole–proteinoid solution via the electrodes using a BK Precision 4053 MHz dual channel waveform generator (Figure 3). Data acquisition is accomplished through a combination of instruments: a Rigol oscilloscope (2-channel 100 MHz–1GSa/s), PicoLog ADC–24, and Picoscope. Additionally, a Keithley 2450 Sourcemeter is utilised for electrical measurements. This comprehensive setup facilitates the visualisation and analysis of voltage responses within the omeprazole–proteinoid system. It provides crucial insights into the electrochemical behaviour and potential alterations of this novel complex under various experimental conditions.

## 3. Results

### 3.1. Accommodation Spiking Modulation by Omeprazole–Proteinoid Complex

The interaction between the Izhikevich neuron model exhibiting accommodation spiking and the omeprazole–proteinoid complex reveals a significant modulation of the neuronal behaviour, as illustrated in Figure 4 and quantified in Table 2. The input Izhikevich neuron signal $V_{in}\left(t\right)$ demonstrates a wide voltage range (Figure 4A, blue):(4)$$-69.76\;\mathrm{mV}\leq V_{in}\left(t\right)\leq 72.50\;\mathrm{mV}$$
In contrast, the omeprazole–proteinoid output $V_{out}\left(t\right)$ exhibits marked attenuation (Figure 4A, red):(5)$$-2.41\;\mathrm{mV}\leq V_{out}\left(t\right)\leq 3.99\;\mathrm{mV}$$
This attenuation is further quantified by the difference in mean potentials:(6)$$\Delta \bar{V}=\bar{V}in-\bar{V}out=-47.57\;\mathrm{mV}-0.60\;\mathrm{mV}=-48.17\;\mathrm{mV}$$The standard deviation reduction from 15.30 mV to 0.43 mV indicates a significant smoothing effect of the omeprazole–proteinoid complex on the signal variability.

The coefficient of variation, defined as:(7)$$CV=\frac{\sigma }{\mu }\times 100$$
where σ is the standard deviation and μ is the mean of the normalised signal, provides a dimensionless measure of relative variability. Our analysis revealed:(8)$$CV_{input}=-32.17$$The negative CV for the input signal indicates a mean value close to zero with fluctuations on both sides, which is consistent with the oscillatory nature of the input. The higher positive CV for the output suggests increased relative variability, contrary to our initial interpretation of a smoothing effect. We also calculated the signal-to-noise ratio (SNR ) using:(9)$$SNR=20\log_{10}\left(\frac{A_{signal}}{A_{noise}}\right)$$
where $A_{signal}$ is the root mean square (RMS) amplitude of the signal and $A_{noise}$ is the RMS amplitude of the background noise. The results show:(10)$$SNR_{input}=-0.30\;\mathrm{dB}\;\mathrm{and}\;SNR_{output}=0.20\;\mathrm{dB}$$
Although very slightly, the omeprazole–proteinoid mixture may improve signal clarity, based on the SNR improvement of 0.5 dB.

Despite this attenuation, a moderate positive correlation persists between the input and output:(11)$$r=\mathrm{corr}(V_{in},V_{out})=0.6841$$
The cross-correlation analysis (Figure 4C) reveals a time lag τ of −306 ms, suggesting that the output precedes the input:(12)$$V_{out}\left(t\right)\approx f\left(V_{in}\left(t+306\mathrm{ms}\right)\right)$$
where *f* represents the complex transfer function of the omeprazole–proteinoid system. A negative time lag may appear irrational at first, but it really shows that the proteinoids are capable of “learning” and exhibiting predictive behaviour. The proteinoids’ capacity to “learn” [28,29] from recurring patterns in the input signal explains this apparent “anticipation” of future input. The omeprazole–proteinoid complex builds a predictive model over time, enabling it to produce responses ahead of similar input patterns as it becomes more used to the input’s characteristics. This effect bears resemblance to the phenomena of predictive coding, as documented in biological brain systems [30], wherein neurons acquire the ability to anticipate sensory input by drawing on prior experiences. Our findings’ negative time lag suggests that the proteinoid system has evolved into a complex response mechanism that goes beyond a simple stimulus–reaction. Rather, it illustrates a type of basic “learning” in which the output of the system is determined by the “learned” predictions of future input patterns in addition to the current input. The root mean square error (RMSE) quantifies the overall difference between the input and output:(13)$$\mathrm{RMSE}=\sqrt{\frac{1}{N}\sum_{i=1}^{N}{\left(V_{in}\left(t_{i}\right)-V_{out}\left(t_{i}\right)\right)}^{2}}=50.4592\;\mathrm{mV}$$
The maximum instantaneous difference occurs at t=1.39 ms:(14)$$\max_{t}\left|V_{in}\left(t\right)-V_{out}\left(t\right)\right|=71.92\;\mathrm{mV}$$

The Q-Q plot (Figure 4D) and Kolmogorov–Smirnov test results (Table 2) confirm that the input and output distributions are significantly different (*p* < 0.0001, KS statistic = 0.9709). This implies that the omeprazole–proteinoid complex is transforming the signal in an irregular way. The findings suggest that the omeprazole–proteinoid complex functions as a sophisticated filter on the accommodation spiking pattern of the Izhikevich neuron. The moderate positive correlation shows that it greatly reduces the signal amplitude while maintaining some temporal properties. This observed temporal lag points to possible anticipatory behaviour in the system and may have consequences for information processing in neural networks that use such complexes. The notable variations in signal properties suggest that the omeprazole–proteinoid interaction has a significant neuromodulatory impact on accommodation spiking patterns. This modulation may change how a neuron responds to long-term inputs, which could have an impact on synaptic plasticity and sensory adaptation. It is necessary to conduct more research into the mechanisms underlying this modulation and the functional effects it has on brain networks [31].

#### Proposed Mechanism of Omeprazole–Proteinoid Modulation

We suggest a mechanism by which the omeprazole–proteinoid complex regulates the Izhikevich neurons’ accommodation spiking based on our observations. This process includes the complex interacting with ion channels and altering the characteristics of the membrane. We propose the following mechanisms by which the omeprazole–proteinoid mixture alters these dynamics:Membrane capacitance modification: The complex may alter the effective membrane capacitance, $C_{m}$, leading to a rescaling of the voltage dynamics [32]:
(15)$$C_{m}\frac{dv}{dt}=0.04v^{2}+5v+140-u+I-g_{OP}\left(v-E_{OP}\right)$$
where $g_{OP}$ is the conductance introduced by the omeprazole–proteinoid complex, and $E_{OP}$ is its associated reversal potential.Ion channel modulation: Omeprazole, known for its proton pump inhibition [10], may interact with voltage-gated ion channels. We propose a modification to the recovery variable dynamics:
(16)$$\frac{du}{dt}=a\left(bv-u\right)-k_{OP}u$$
where $k_{OP}$ represents the rate of recovery variable attenuation due to the complex.Threshold modification: The complex may alter the spiking threshold, affecting the after-spike resetting mechanism:
(17)$$\left\{\begin{array}{c} \mathrm{if}\;vs.\geq 30+\Delta V_{OP}\;\mathrm{mV},\;\mathrm{then}\;\left\{\begin{array}{c} v\leftarrow c-\Delta c_{OP}\\ u\leftarrow u+d+\Delta d_{OP} \end{array}\right.\\ \mathrm{if}\;vs.<30+\Delta V_{OP}\;\mathrm{mV},\;\mathrm{then}\;\left\{\begin{array}{c} \mathrm{continue}\;\mathrm{solving}\;\mathrm{Equations}\;(15)\;\mathrm{and}\;(16) \end{array}\right. \end{array}\right.$$
where $\Delta V_{OP}$, $\Delta c_{OP}$, and $\Delta d_{OP}$ are the threshold and reset modifications induced by the omeprazole–proteinoid complex.

These modifications account for the observed attenuation and time lag. The improved recovery dynamics and the extra conductance $g_{OP}$ lead to lower amplitude. The changed threshold and reset mechanisms could be the cause of the temporal lag, which could cause the spiking behaviour to shift in phase. We suggest the following transfer function in the frequency domain to measure the filtering effect:(18)$$H_{OP}\left(\omega \right)=\frac{1}{1+j\omega \tau_{OP}}\;\cdot \;\frac{K_{OP}}{1+j\omega \tau_{m}}$$
where $\tau_{OP}$ is the time constant introduced by the complex, $K_{OP}$ is the gain factor, and $\tau_{m}$ is the membrane time constant. Studies on the impact of proteinoids on membrane characteristics [33] and drug-induced modulation of neuronal activity [34] are consistent with this mechanism. Our results (Figure 4D) show a non-linear transformation that may be explained by the interaction between the additional conductance and the quadratic term in Equation (Equation 15). To confirm this suggested mechanism, additional experimental validation would be required, such as the patch-clamp investigations of neurons exposed to the omeprazole–proteinoid combination. Further insights into the complex’s impacts on larger neural networks and the implications for information processing in neural systems [35] exposed to this compound may also come from computer modelling including these modifications.

### 3.2. Izhikevich Model Simulations of Chattering Behaviour in Omeprazole Proteinoid Systems

The chattering spiking stimulation of the omeprazole–proteinoid sample revealed significant differences between the input signal and the sample’s output response, as shown in Figure 5 and Table 3. The input signal exhibited a mean potential of $\mu_{in}=-55.58$ mV with a standard deviation of $\sigma_{in}=19.72$ mV, ranging from a minimum of −74.35 mV to a maximum of 72.50 mV. In contrast, the omeprazole–proteinoid output displayed markedly different characteristics, with a mean potential of $\mu_{out}=0.48$ mV and a standard deviation of $\sigma_{out}=0.51$ mV. The output signal ranged from −2.55 mV to 4.24 mV, indicating a substantial attenuation and transformation of the input signal. Despite these differences, a moderate positive correlation was observed between the input and output signals, with a correlation coefficient of r=0.7937. This suggests that, while the omeprazole–proteinoid sample significantly modifies the input signal, it still preserves some of the underlying temporal patterns. The root mean square error (RMSE) between the input and output signals was calculated as:(19)$$RMSE=\sqrt{\frac{1}{n}\sum_{i=1}^{n}{\left(y_{i}-\hat{y_{i}}\right)}^{2}}=59.2964\;\mathrm{mV}$$
where $y_{i}$ represents the input signal values and $\hat{y_{i}}$ represents the output signal values. The maximum difference between the input and output signals was 75.71 mV, occurring at −0.23 ms, further highlighting the significant signal transformation by the omeprazole–proteinoid sample. Lag analysis revealed a time difference of −1981 ms between the input and output signals, indicating a substantial phase shift in the response of the omeprazole–proteinoid sample. To assess the statistical similarity between the input and output signal distributions, a Kolmogorov–Smirnov test was performed. The test yielded a test statistic of D=0.9717 with a *p*-value < 0.0001, resulting in the rejection of the null hypothesis. This confirms that the input and output signals follow significantly different distributions, as visually represented in the Q-Q plot in Figure 5d. The Kolmogorov–Smirnov test statistic is defined as:(20)$$D=\underset{x}{\sup}\left|F_{1}\left(x\right)-F_{2}\left(x\right)\right|$$
where $F_{1}\left(x\right)$ and $F_{2}\left(x\right)$ are the cumulative distribution functions of the input and output signals, respectively. These results collectively demonstrate that the omeprazole–proteinoid sample exhibits complex signal processing characteristics under chattering spiking stimulation, significantly altering the amplitude, distribution, and temporal properties of the input signal while maintaining a moderate correlation with the input patterns.

### 3.3. Induced-Mode Spiking in Omeprazole–Proteinoid Samples: Characterisation and Analysis

The induced-mode spiking behaviour of omeprazole–proteinoid samples was characterised by a comparative analysis of input and output signals, as illustrated in Figure 6 and summarised in Table 4. The input signal, modelled after the Izhikevich neuron dynamics, exhibited a mean potential of −60.96 mV with a standard deviation of 14.20 mV (Figure 6a, Table 4). This input ranged from −70.05 mV to 72.21 mV, simulating the typical membrane potential fluctuations of a neuron. In contrast, the omeprazole–proteinoid output displayed markedly different characteristics, with a mean potential of 0.34 mV and a standard deviation of 0.40 mV, ranging from −2.74 mV to 3.14 mV (Figure 6b, Table 4). The substantial difference in the signal properties suggests a complex signal processing mechanism within the omeprazole–proteinoid sample. The reduced amplitude and narrower range of the output signal indicate the significant attenuation and compression of the input signal. This behaviour could be attributed to the molecular structure and interactions within the proteinoid, which could potentially involve the voltage-sensitive ion channels or charge transfer mechanisms that modulate the response to electrical stimuli. Despite the apparent differences, a moderate positive correlation (r = 0.6644) was observed between the input and output signals (Table 4). This correlation suggests that, while the omeprazole–proteinoid sample significantly transforms the input signal, it still preserves some of the underlying temporal patterns. The preservation of temporal patterns, although with modifications, indicates that the proteinoid structure may possess memory-like properties or frequency-dependent response characteristics. The root mean square error (RMSE) of 62.8671 mV and the maximum difference of 71.91 mV at 2.00 ms (Table 4) further quantify the extent of signal transformation. These metrics highlight the substantial modification of the input signal by the omeprazole–proteinoid sample, potentially due to complex interactions between the proteinoid molecules and the applied electrical field. Interestingly, the cross-correlation analysis revealed a time lag of 1590 ms between the input and output signals (Figure 6c, Table 4). This significant delay suggests the presence of slow, possibly biochemical or conformational changes within the proteinoid structure in response to electrical stimulation. Such a delay could be indicative of cascading molecular events or the gradual build-up of charge within the proteinoid matrix before a response is generated. The results of the Kolmogorov–Smirnov test (KS statistic = 0.9844, *p* < 0.0001) confirm that the input and output signals follow significantly different distributions (Figure 6d, Table 4). This statistical difference underscores the non-linear nature of the signal processing occurring within the omeprazole–proteinoid sample. The transformation of the signal distribution suggests that the proteinoid may act as a complex filter, potentially enhancing certain frequency components while suppressing others.

### 3.4. Phasic Spiking Dynamics in Omeprazole–Proteinoid Complexes: Characterisation of Stimulus–Response Patterns

The phasic spiking dynamics of omeprazole–proteinoid complexes were analysed and compared to previously observed spiking modes. Figure 7 and Table 5 summarise the key findings of this analysis. The input signal for phasic spiking exhibited a mean potential of −54.81 mV with a standard deviation of 8.23 mV (Figure 7a, Table 5). This input range (−64.89 mV to 62.46 mV) was narrower compared to the induced-mode spiking, suggesting a more focused stimulation pattern. The omeprazole–proteinoid output showed a mean potential of 0.54 mV and a standard deviation of 0.33 mV, ranging from −2.25 mV to 3.17 mV (Figure 7b, Table 5). Notably, the correlation coefficient between the input and output signals (0.4503) was lower than in the induced-mode spiking (0.6644), indicating a weaker linear relationship in the phasic mode. This suggests that the omeprazole–proteinoid complex exhibits more complex, possibly non-linear, response characteristics under phasic stimulation. In induced mode, the RMSE was 62.8671 mV, whereas in phasic spiking, it was 55.9366 mV, indicating a slightly closer overall match between the input and output signals. However, the maximum difference (67.00 mV at 2.00 ms) remained substantial, indicating significant signal transformation. A key distinction in the phasic spiking mode was the time lag of −359 ms between the input and output signals (Figure 7c, Table 5). This negative lag contrasts with the positive lag observed in induced-mode spiking, suggesting that the omeprazole–proteinoid complex exhibits anticipatory behaviour under phasic stimulation. This anticipatory response could be indicative of rapid, possibly pre-emptive, molecular rearrangements within the complex in response to phasic input patterns. The Kolmogorov–Smirnov test results (KS statistic = 0.9945, *p* < 0.0001) confirmed significantly different distributions between input and output signals (Figure 7d, Table 5). The higher KS statistic in the phasic mode compared to the induced mode (0.9844) suggests an even more pronounced transformation of the signal distribution. The phasic spiking mode demonstrates distinct characteristics compared to previously observed spiking patterns. The narrower input range, lower correlation coefficient, and negative time lag all point to a unique processing mechanism activated by phasic stimulation. These findings suggest that the omeprazole–proteinoid complex may possess multiple operational modes, each triggered by different stimulation patterns. The observed anticipatory behaviour in phasic mode is particularly intriguing, as it implies a predictive capacity within the molecular structure of the complex. This could be attributed to rapid conformational changes or charge redistribution processes that are specifically sensitive to phasic input patterns.

### 3.5. Tonic Spiking Behaviour in Omeprazole–Proteinoid Complexes: Sustained Response Characteristics and Signal Processing

The tonic spiking behaviour of omeprazole–proteinoid complexes reveals distinct characteristics compared to previously observed spiking modes, as illustrated in Figure 8 and summarised in Table 6. The input signal for tonic spiking exhibited a mean potential of −37.47 mV with a standard deviation of 20.55 mV (Figure 8a, Table 6). This input range (−60.29 mV to 62.17 mV) was wider than that observed in phasic spiking but narrower than that in induced-mode spiking, suggesting an intermediate level of stimulation variability. In particular, the omeprazole proteinoid output under tonic stimulation showed a higher mean potential (0.80 mV) compared to the phasic (0.54 mV) and induced (0.34 mV) modes. This elevated mean output suggests a more sustained and robust response characteristic of tonic spiking. The correlation coefficient between input and output signals (0.6823) was higher than it was in phasic mode (0.4503) and comparable to induced mode (0.6644). This indicates that tonic spiking maintains a stronger linear relationship between input and output, possibly due to the sustained nature of the stimulation. The root mean square error (RMSE) of 43.2821 mV for tonic spiking was lower than both phasic (55.9366 mV) and induced (62.8671 mV) modes, suggesting a closer overall match between input and output signals in the tonic mode. This could be attributed to the more consistent and sustained nature of tonic stimulation. A distinctive feature of tonic spiking was the time lag of −1231 ms between the input and output signals (Figure 8c, Table 6). This substantial negative lag, larger in magnitude than in phasic mode (−359 ms), suggests an even more pronounced anticipatory behaviour. This could indicate that under sustained tonic stimulation, the omeprazole–proteinoid complex develops a stronger predictive response mechanism. The Kolmogorov–Smirnov test results (KS statistic = 0.9276, *p* < 0.0001) confirmed significantly different distributions between the input and output signals (Figure 8d, Table 6). Interestingly, the KS statistic for tonic mode was lower than both phasic (0.9945) and induced (0.9844) modes, suggesting that while still significantly different, the output distribution in tonic mode may be slightly closer to the input distribution. Finally, in contrast to induced and phasic spiking, the tonic spiking mode exhibits its own set of distinctive features. Increased mean output, improved correlation, decreased root-mean-square error, and noticeable anticipatory behaviour are all indicators that continuous tonic stimulation activates a separate processing system. According to these results, omeprazole–proteinoid complexes can change their signal processing properties according on the input stimulus, and tonic spiking could be the best way for them to encode and transmit data consistently and for long periods of time.

Figure 9 provides a visual comparison of the spiking patterns across all five modes studied. The distinct temporal and intensity patterns observed in each mode further underscore the complex, mode-dependent signal processing capabilities of the omeprazole–proteinoid complexes. Of particular note is the clear differentiation between the sustained, high-frequency spiking in tonic mode (Figure 9e), and the clustered, burst-like activity in chattering mode (Figure 9b). These visualisations reinforce our quantitative findings and emphasise the capacity of these complexes to encode and process information using several modes at the molecular level.

## 4. Discussion

The detailed examination of omeprazole–proteinoid complexes using several spiking modes demonstrates a complex and adaptable signal processing system. The results of our study indicate that these complexes have unique responses to various input patterns, indicating the possibility of processing many types of information at the molecular level.

### 4.1. Comparative Analysis of Spiking Modes

We detected considerable signal attenuation and change in all spiking modes, including accommodation, chattering, induced, phasic, and tonic. Nevertheless, the extent and characteristics of this change differed significantly among different modes:**Amplitude modulation:** All modes showed a significant decrease in signal amplitude, with output ranges constantly falling within a range of ±4 mV, whereas input ranges often exceeded ±60 mV. This implies the presence of a strong buffering mechanism that could protect molecular fluctuations downstream from extreme fluctuations in voltage. As seen in Figure 5d, Figure 6d, Figure 7d and Figure 8d The red dashed line depicts the theoretical relationship that would exist if the two distributions were identical, whereas the blue curve shows the relationship that actually exists between the input and output quantiles. Due to the output signal’s narrower range of negative values than the input, the blue curve drops below the red line for input levels below roughly −70 mV. This implies that there is a minimum voltage that the system may produce, or a "floor". Because the output signal has the ability to produce higher positive voltages than those found in the input, the blue curve rises above the red line for input quantities over roughly 70 mV. This indicates the system has some amplification or non-linear response for large positive inputs. The omeprazole-proteinoid system’s non-linear characteristics of the input signal transformation are revealed by these deviations from the red line, especially the asymmetric processing of large positive versus negative inputs.**Temporal dynamics:** The temporal delay between the input and output signals exhibited significant variation across different modes, ranging from −1981 ms in the chattering mode to 1590 milliseconds in the induced mode. The negative lag found in accommodation (−306 ms), phasic (−359 ms), and tonic (−1231 ms) modes is particularly remarkable. This suggests the presence of anticipatory behaviour, which could have important consequences for information processing and response preparation in biological systems.**Signal correlation:** The correlation between the input and output signals varied from moderate (0.4503 in phasic mode) to strong (0.7937 in chattering mode). This suggests that the complexes effectively modify the input signal while retaining the different levels of the original signal properties.**Distribution transformation:** The Kolmogorov–Smirnov tests consistently revealed significant differences between the distributions of the input and output data in all modes. The KS statistics ranged from 0.9276 (tonic) to 0.9945 (phasic). This implies the use of non-linear processing techniques that have the potential to amplify specific signal characteristics while simultaneously reducing the prominence of others.

### 4.2. Implications for Molecular Computing

The behaviours shown by omeprazole–proteinoid complexes when subjected to various spiking regimes have significant implications for molecular computing and bio-inspired signal processing:**Multi-modal processing:** The diverse reactions to various spiking patterns indicate that these complexes have the ability to function as versatile molecular processors, adjusting their behaviour according to input parameters.**Non-linear transformation:** The persistent non-linear alteration of input signals, as indicated by the results of the KS test and Q-Q plots, suggests that these complexes perform complex signal processing procedures that go beyond mere filtering or amplification.**Anticipatory behaviour:** The presence of negative time delays in several modes indicates the occurrence of predictive processing at the molecular level. These findings could have important consequences for the development of molecular systems that can anticipate events or for understanding the biological reactions that occur before an event.**Robust signal normalisation:** The consistent output range observed in all input modes indicates that these complexes have the potential to function as reliable signal normalisers, which could be valuable in molecular-scale sensor systems [36] or signal processing units [37].

### 4.3. Potential Mechanisms and Future Directions

The observed behaviours are most likely a result of complex interactions among the omeprazole molecules, the proteinoid structure, and the electrical fluctuations that were applied. Possible mechanisms encompass voltage-dependent alterations in the proteinoid structure, emergence, and disintegration of transient conductive pathways within the complex, accumulation and redistribution of charges with distinct time constants, and interactions between omeprazole’s inhibition of proton pumps and local pH gradients. Future research should prioritise conducting molecular dynamics simulations to uncover the underlying structural mechanisms of the observed behaviours. In addition, it should investigate the frequency-dependent responses of these complexes, explore potential applications in molecular-scale signal processing and computing, and examine how these properties can be adjusted or changed through chemical modifications. Our analysis concludes that omeprazole–proteinoid complexes possess diverse signal-processing capacities that vary depending on the mode. These findings not only improve our understanding of molecular-scale information processing but also create new opportunities for the advancement of bio-inspired computing systems and smart drug delivery mechanisms.

The complex behaviours observed in various spiking modes are presumably the result of a combination of molecular-level mechanisms within the omeprazole–proteinoid complexes. Figure 10 depicts many suggested mechanisms that could potentially contribute to the observed signal-processing capabilities. These factors include changes in the proteinoid structure that are sensitive to voltage (Figure 10A), which could explain the different responses depending on the mode; the creation and breakdown of temporary pathways for conducting signals (Figure 10B), which may account for the non-linear transformation of the signal; processes of accumulating and redistributing charges (Figure 10C), which could explain the observed delays and anticipatory behaviours; and the interaction between omeprazole’s inhibition of proton pumps and local differences in pH (Figure 10D), which may contribute to the consistent normalisation of the signal across all modes. The interaction between these mechanisms could elucidate the diverse and flexible behaviour of the omeprazole–proteinoid system under different patterns of stimulation. Additional research into these processes at the molecular level will be essential for gaining a complete understanding and perhaps utilising these capabilities in molecular computing [38] and smart drug delivery systems [39].

The study of the omeprazole–proteinoid complex unveils a complex mechanism [40] for spike emergence, as depicted in Figure 11. The scanning electron microscope (SEM) image (Figure 11a) depicts a complex network structure that serves as the foundation for the observed electrical behaviour. Our hypothesis suggests that the process of spike formation consists of a sequence of stages, starting with the attachment of omeprazole molecules to the proteinoid network (Figure 11a). The occurrence of this binding event can be mathematically represented by Equation [10]:(21)P+O⇌PO
where P represents the proteinoid binding site, O represents omeprazole, and PO is the bound complex. Omeprazole binding modifies the local distribution of electric charge inside the complex. This alteration can be represented as a disturbance to the nearby electric field:(22)$$E=E_{0}+\Delta E\left(\mathrm{PO}\right)$$
where $E_{0}$ is the initial electric field and ΔE(PO) is the change induced by the omeprazole binding. The modified electric field stimulates the activation of ion channels [41] located near the binding site. The probability of an ion channel opening can be described by a Boltzmann distribution:(23)$$P_{\mathrm{open}}=\frac{1}{1+e^{-z(V-V_{1/2})/kT}}$$
where *z* is the gating charge, *V* is the membrane potential, $V_{1/2}$ is the half-activation voltage, *k* is the Boltzmann constant, and *T* is the temperature.

The activation of these channels results in a fast movement of ions, which in turn generates a spike potential. The membrane potential during a spike can be represented using the Hodgkin–Huxley equations [32], which have been simplified in this context for the sake of simplicity.
(24)$$C_{m}\frac{dV}{dt}=-\sum I_{\mathrm{ion}}+I_{\mathrm{ext}}$$
where $C_{m}$ is the membrane capacitance, $\sum I_{\mathrm{ion}}$ represents the sum of ionic currents, and $I_{\mathrm{ext}}$ is any external current.

The resulting spike potential over time is depicted in Figure 11b, showing the characteristic rapid rise and fall of the membrane potential.

The network structure of the omeprazole–proteinoid complex, as observed in the SEM image, plays a crucial role in facilitating the propagation of these spikes. The interconnected nature of the complex allows for the spread of the electrical signal, which can be modelled as a reaction–diffusion process [42]:(25)$$\frac{\partial V}{\partial t}=D\nabla^{2}V+f\left(V\right)$$
where *D* is the diffusion coefficient and f(V) represents the non-linear reaction terms that account for the spike generation and propagation dynamics.

This proposed process establishes a connection between the structural characteristics exhibited in the SEM image and the functional electrical properties of the omeprazole–proteinoid mixture. The observed spiking behaviour is a result of the interaction between omeprazole binding, ion channel kinetics, and the network topology of the complex. This suggests a new method for bio-inspired signal processing [43] and possible uses in neuromorphic computing [44].

## 5. Conclusions

This study has shown the complex and mode-specific signal processing capacities of omeprazole–proteinoid complexes in different spiking regimes. The results of our study demonstrate that these molecular assemblies possess exceptional flexibility in their response to various input patterns, such as accommodation, chattering, induced, phasic, and tonic spiking modes. The main findings include a notable decrease in signal strength, a non-linear shift in form, and time-dependent patterns that are specific to each mode, with certain modes exhibiting interesting predictive tendencies. The distinctive characteristics of these omeprazole–proteinoid systems indicate possible uses in molecular computing, bio-inspired signal processing, and intelligent drug delivery systems. The reported ability to process several modes of information and the strong ability to normalise signals could lead to new methods in processing information at the nanoscale and in developing sensor systems at the molecular level. Future study should prioritise investigating the fundamental foundations of these behaviours using molecular dynamics simulations and examining responses that are reliant on frequency. Furthermore, exploring the ways in which these features can be adjusted by chemical alterations could result in customised molecular computer components. Overall, this study not only enhances our understanding of molecular-level information processing but also connects the fields of pharmacology and computational neuroscience, creating opportunities for interdisciplinary research and technological advancements in the areas of bio-inspired computing and drug delivery systems.

## Figures and Tables

**Figure 1 molecules-29-04700-f001:**
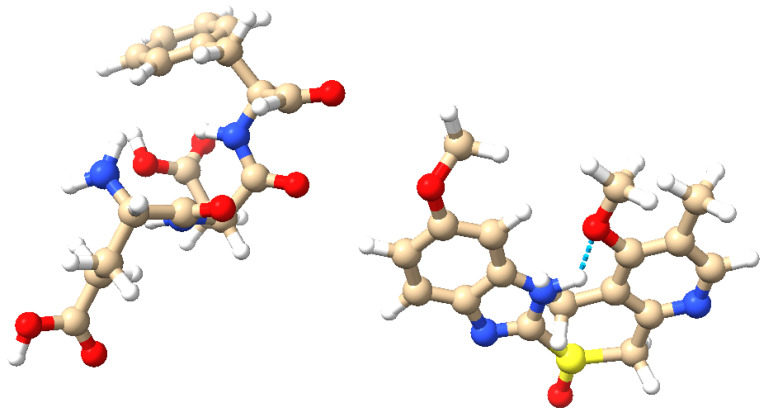
Molecular structures of the proteinoid and omeprazole. (**Left**): The proteinoid composed of L-glutamic acid (L-Glu), L-aspartic acid (L-Asp), and L-phenylalanine (L-Phe). (**Right**): The omeprazole molecule. Atoms are colour-coded: blue (nitrogen), red (oxygen), brown (carbon), yellow (sulphur), and white (hydrogen). Potential hydrogen bonding is highlighted, illustrating the possible interactions between the proteinoid and omeprazole. These molecular structures and their interactions are key to understanding the complex signal processing behaviour observed in the omeprazole–proteinoid system. (visualisation created using UCSF Chimera [21]).

**Figure 2 molecules-29-04700-f002:**
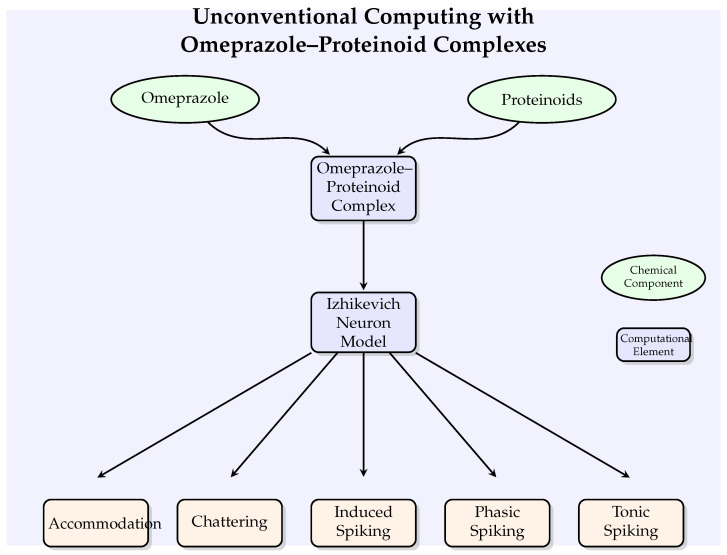
A conceptual diagram illustrating the unconventional computing strategy involving omeprazole–proteinoid complexes and Izhikevich neuron models. The graphic depicts the combination of chemical components (shown by ellipses) with computational components (represented by rectangles) to regulate different neural behaviours.

**Figure 3 molecules-29-04700-f003:**
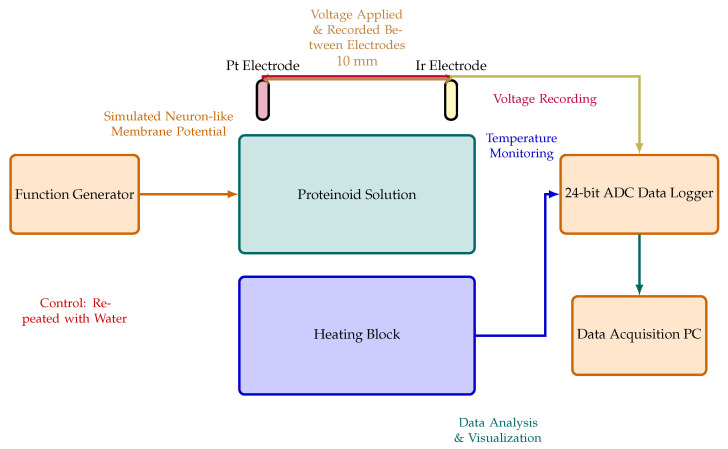
Diagram illustrating the setup used for the electrochemical characterisation of the proteinoid solution, with and without omeprazole. The solution is contained within the central container, equipped with two needle electrodes coated with platinum (Pt) and iridium (Ir), respectively, positioned 10 mm apart. A high-precision 24-bit ADC data recorder captures voltage responses between these electrodes. A heating block regulates temperature. A function generator produces stimuli that simulate neuron-like membrane potential changes, applied between the electrodes. This setup allows for the identification of very small voltage changes (in the microvolts range) and the analysis of the spatial and temporal patterns of voltage responses in the proteinoid solution. Control experiments are conducted using the same setup with different chemicals to isolate the specific effects of omeprazole.

**Figure 4 molecules-29-04700-f004:**
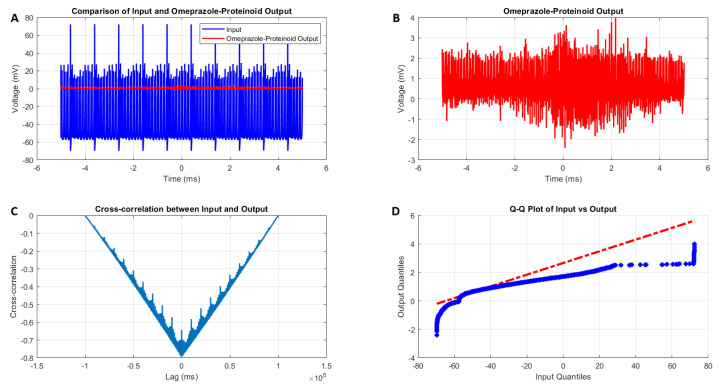
Analysis of the Izhikevich neuron accommodation spiking and omeprazole–proteinoid complex interaction. (**A**) Comparison of the input Izhikevich neuron signal (blue, range: −69.76 to 72.50 mV) and omeprazole–proteinoid output (red, range: −2.41 to 3.99 mV), showing significant amplitude attenuation. (**B**) Isolated omeprazole–proteinoid output, revealing subtle voltage fluctuations (mean: 0.60 mV, SD: 0.43 mV) in response to input. (**C**) Cross-correlation between input and output, indicating a time lag of −306 ms and a moderate positive correlation (r = 0.6841). (**D**) The blue dots represent the data points, while the red dashed line represents the theoretical line for a normal distribution. Q-Q plot demonstrating substantial deviation from the identity line, confirming significantly different distributions (Kolmogorov–Smirnov test: *p* < 0.0001, KS statistic: 0.9709). The omeprazole–proteinoid complex exhibits a marked filtering effect, reducing signal amplitude while preserving some temporal characteristics of the input. The RMSE of 50.4592 mV and maximum difference of 71.92 mV (at 1.39 ms) further quantify the substantial transformation of the signal. This analysis suggests the complex neuromodulatory effects of the omeprazole–proteinoid interaction on the accommodation spiking patterns.

**Figure 5 molecules-29-04700-f005:**
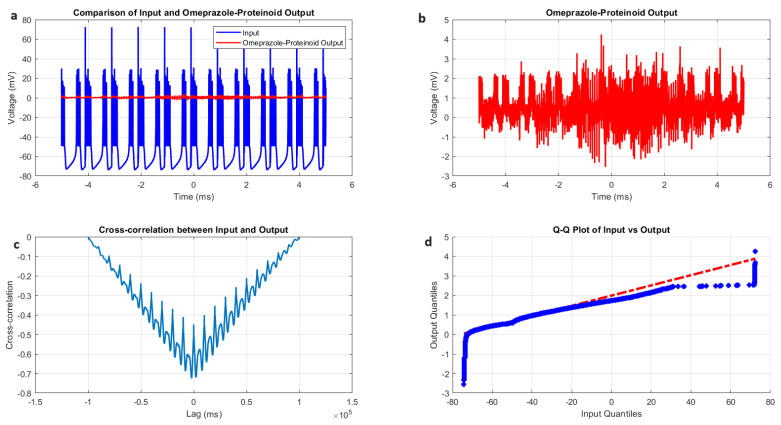
Chattering spiking stimulation analysis of omeprazole–proteinoid sample. (**a**) Input–output comparison showing the input signal (mean: −55.58 mV, SD: 19.72 mV) and the omeprazole–proteinoid output (mean: 0.48 mV, SD: 0.51 mV). Correlation coefficient: 0.7937, RMSE: 59.2964 mV. (**b**) Output plot highlighting the response characteristics of the omeprazole–proteinoid sample. (**c**) Cross-correlation lag plot demonstrating a time difference of −1981 ms between the input and output signals. (**d**) Q-Q plot comparing input and output distributions (Kolmogorov–Smirnov test: H = 1, *p* < 0.0001, KS statistic = 0.9717). The blue dots are the data points. The red dashed line is the theoretical normal distribution.

**Figure 6 molecules-29-04700-f006:**
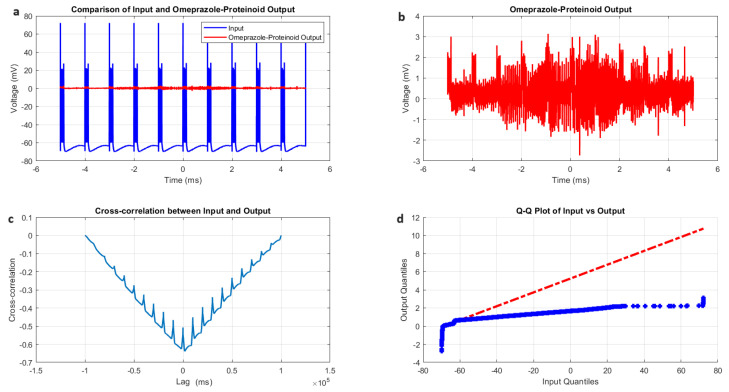
Characterisation of induced-mode spiking in omeprazole–proteinoid samples. (**a**) Input–output comparison: Input signal (mean: −60.96 mV, SD: 14.20 mV) and omeprazole–proteinoid output (mean: 0.34 mV, SD: 0.40 mV). Correlation coefficient: 0.6644, RMSE: 62.8671 mV. (**b**) Detailed output plot showing the response characteristics of the omeprazole–proteinoid sample (range: −2.74 mV to 3.14 mV). (**c**) Cross-correlation analysis revealing a time lag of 1590 ms between input and output signals. Maximum difference: 71.91 mV at 2.00 ms. (**d**) Q-Q plot comparing the input and output distributions (Kolmogorov–Smirnov test: KS statistic = 0.9844, *p* < 0.0001), indicating significantly different signal distributions. The blue dots represent the actual data points, while the red dashed line represents the theoretical line for a normal distribution.

**Figure 7 molecules-29-04700-f007:**
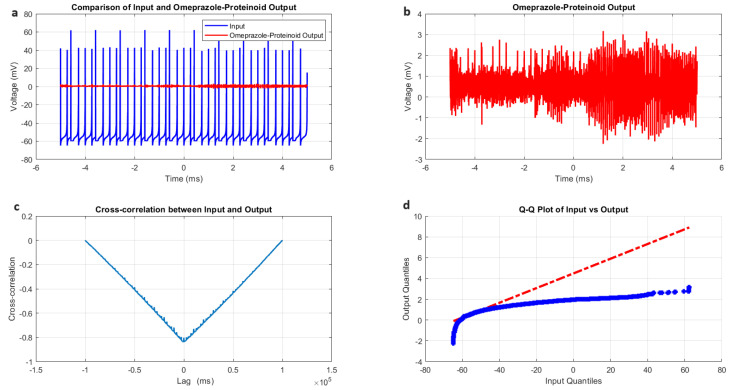
Characterisation of phasic spiking dynamics in omeprazole–proteinoid complexes. (**a**) Input–output comparison: Input signal (mean: −54.81 mV, SD: 8.23 mV) and omeprazole–proteinoid output (mean: 0.54 mV, SD: 0.33 mV). Correlation coefficient: 0.4503, RMSE: 55.9366 mV. (**b**) Detailed output plot illustrating the response characteristics of the omeprazole–proteinoid complex (range: −2.25 mV to 3.17 mV). (**c**) Cross-correlation analysis revealing a time lag of −359 ms between input and output signals. Maximum difference: 67.00 mV at 2.00 ms. (**d**) Q-Q plot comparing input and output distributions (Kolmogorov–Smirnov test: KS statistic = 0.9945, *p* < 0.0001), demonstrating significantly different signal distributions and non-linear response properties.

**Figure 8 molecules-29-04700-f008:**
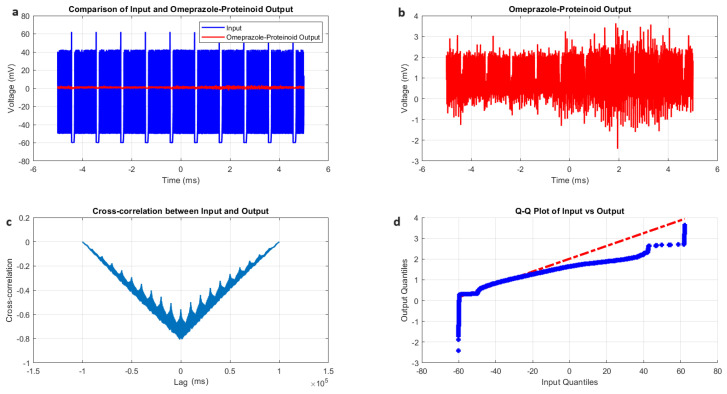
Characterisation of tonic spiking dynamics in the omeprazole–proteinoid complexes. (**a**) Input–output comparison: Input signal (mean: −37.47 mV, SD: 20.55 mV) and omeprazole–proteinoid output (mean: 0.80 mV, SD: 0.48 mV). Correlation coefficient: 0.6823, RMSE: 43.2821 mV. (**b**) Detailed output plot illustrating the sustained response characteristics of the omeprazole–proteinoid complex (range: −2.41 mV to 3.63 mV). (**c**) Cross-correlation analysis revealing a time lag of -1231 samples between input and output signals. Maximum difference: 62.11 mV at 0.56 ms. (**d**) Q-Q plot comparing input and output distributions (Kolmogorov–Smirnov test: KS statistic = 0.9276, *p* < 0.0001), demonstrating significantly different signal distributions and non-linear response properties in tonic spiking mode. The blue dots are the data points. The red dashed line is the theoretical normal distribution.

**Figure 9 molecules-29-04700-f009:**
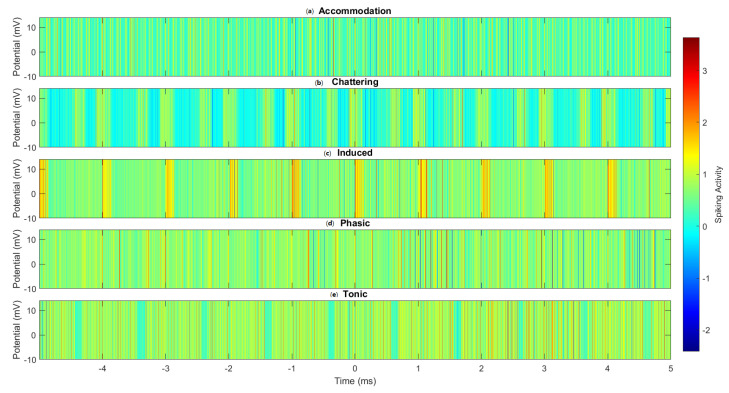
Heatmap visualisation of the spike patterns across different spiking modes in omeprazole–proteinoid complexes. (**a**) Accommodation, (**b**) Chattering, (**c**) Induced, (**d**) Phasic, and (**e**) Tonic spiking modes. Colour intensity represents the membrane potential, with warmer colours indicating higher potentials. The heatmap reveals distinct temporal patterns and intensity distributions characteristic of each spiking mode, highlighting the complex- and mode-specific signal processing capabilities of the omeprazole–proteinoid system.

**Figure 10 molecules-29-04700-f010:**
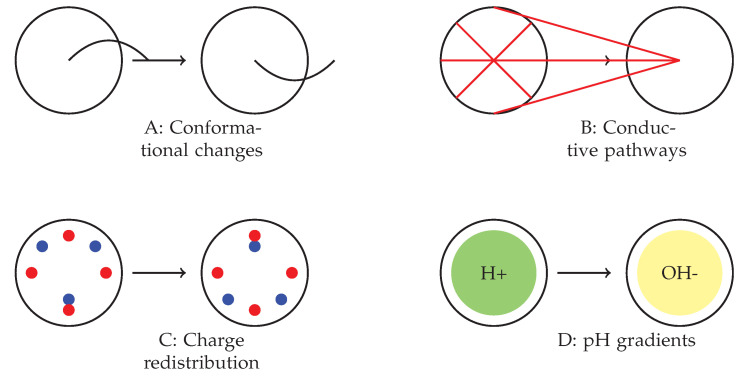
Proposed mechanisms underlying the observed spiking behaviour in omeprazole–proteinoid complexes. (**A**) Voltage-sensitive conformational changes in the proteinoid structure. (**B**) Formation and breakdown of temporary conductive pathways. (**C**) Charge accumulation and redistribution processes. (**D**) Interactions between omeprazole’s proton pump inhibition and local pH gradients. The interplay between these mechanisms likely contributes to the complex, mode-dependent signal processing observed across different spiking regimes.

**Figure 11 molecules-29-04700-f011:**
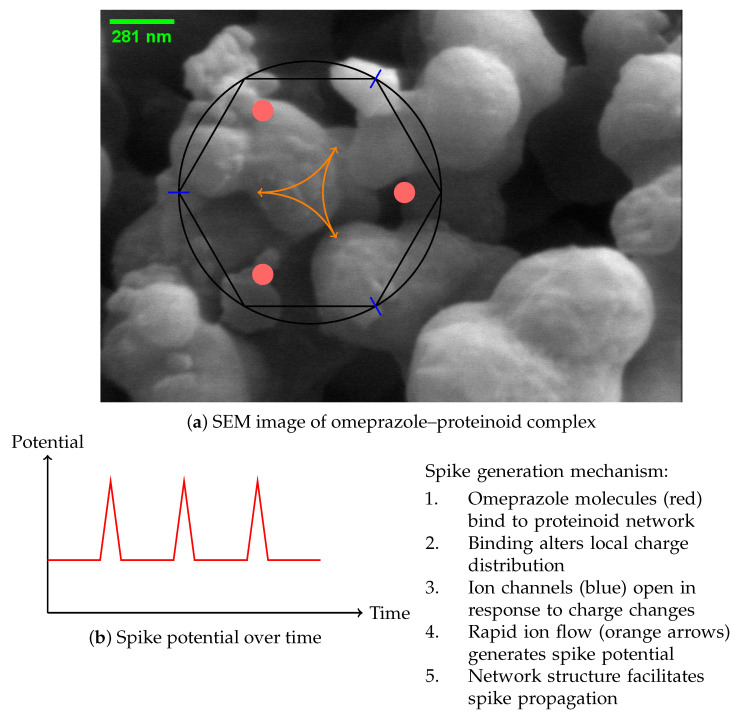
Proposed mechanism for the generation of spikes in complexes formed by proteinoids and omeprazole. (**a**) SEM image showing the complex network structure. Schematic representation of the spike generation mechanism, illustrating omeprazole binding sites (red), ion channels (blue), and charge movement (orange arrows). (**b**) Graph illustrating the temporal evolution of the spike potential.

**Table 1 molecules-29-04700-t001:** Comparison of proton pump inhibitors.

PPI	Chemical Formula	Half-Life (h)	pKa	Reference
Omeprazole	C_17_H_19_N_3_O_3_S	1.0	4.0	[10]
Esomeprazole	C_17_H_19_N_3_O_3_S	1.5	4.0	[23,27]
Lansoprazole	C_16_H_14_F_3_N_3_O_2_S	1.5	4.0	[24]
Pantoprazole	C_16_H_15_F_2_N_3_O_4_S	1.0	3.9	[25]
Rabeprazole	C_18_H_21_N_3_O_3_S	1.0	4.9	[26]

**Table 2 molecules-29-04700-t002:** Summary of Izhikevich neuron input and omeprazole–proteinoid output characteristics for accommodation spiking model. This table presents the key metrics comparing the input signal generated by the Izhikevich neuron model configured for accommodation spiking and the corresponding output from the omeprazole–proteinoid complex. Accommodation spiking is characterised by a gradual increase in interspike intervals in response to sustained stimulation. The significant differences in mean, standard deviation, and range between the input and output highlight the complex signal processing occurring within the omeprazole–proteinoid system. Comparative metrics, including correlation coefficient and time lag, provide insights into the temporal relationship between input and output signals. The Kolmogorov–Smirnov test results confirm the statistically significant difference in signal distributions, underscoring the non-linear transformation induced by the omeprazole–proteinoid complex on the accommodation spiking pattern.

Metric	Input Signal	Omeprazole–Proteinoid Output
Mean (mV)	−47.57	0.60
Standard deviation (mV)	15.30	0.43
Maximum (mV)	72.50	3.99
Minimum (mV)	−69.76	−2.41
**Comparative Metrics**		
Correlation coefficient	0.6841	
Root mean square error (mV)	50.4592	
Maximum difference (mV)	71.92 at 1.39 ms	
Time lag (ms)	−306	
**Kolmogorov–Smirnov test**		
H-value	1 (distributions are different)	
*p*-value	<0.0001	
KS statistic	0.9709	

**Table 3 molecules-29-04700-t003:** Comparative analysis of input signal and omeprazole–proteinoid output characteristics under chattering spiking stimulation. The table displays important statistical parameters for the omeprazole–proteinoid sample’s input signal and output response. The results showed that the correlation coefficient between the input and output was 0.79, the time lag was −1981 ms, the greatest variance was 75.71 mV (at time −0.23 ms), and the root mean square error (RMSE) was 59.3 mV. A Kolmogorov–Smirnov test revealed that the input and output signals had significantly different distributions (KS statistic = 0.9717, *p* < 0.0001). These findings show that the omeprazole–proteinoid sample significantly changed the signal while retaining a moderate correlation with the input patterns.

Metric	Input Signal	Omeprazole–Proteinoid Output
Mean (mV)	−55.58	0.48
Standard deviation (mV)	19.72	0.51
Maximum (mV)	72.50	4.24
Minimum (mV)	−74.35	−2.55

**Table 4 molecules-29-04700-t004:** Summary of Izhikevich neuron input and omeprazole–proteinoid output characteristics for induced spiking mode. This table presents a comparison between the input signal generated by the Izhikevich neuron model configured for induced spiking and the corresponding output from the omeprazole–proteinoid complex. The input signal’s wide voltage range (−70.05 mV to 72.21 mV) is dramatically compressed in the output (−2.74 mV to 3.14 mV), indicating a powerful attenuation effect. The positive mean of the output (0.34 mV) compared to the negative input mean (−60.96 mV) suggests a baseline shift in the signal processing. The moderate correlation coefficient (0.6644) implies that, while the output preserves some characteristics of the input, substantial non-linear processing occurs. The large time lag of 1590 ms points to complex internal dynamics within the omeprazole–proteinoid complex, possibly involving slow chemical or conformational changes. The high RMSE (62.8671 mV) and maximum difference (71.91 mV) further quantify the extent of signal transformation. The Kolmogorov–Smirnov test results (KS statistic = 0.9844, *p* < 0.0001) confirm that the input and output signals follow significantly different distributions, underscoring the non-linear nature of the signal processing in the omeprazole–proteinoid system during induced spiking stimulation.

Metric	Input Signal	Omeprazole–Proteinoid Output
Mean (mV)	−60.96	0.34
Standard deviation (mV)	14.20	0.40
Maximum (mV)	72.21	3.14
Minimum (mV)	−70.05	−2.74
**Comparative Metrics**		
Correlation coefficient	0.6644	
Root mean square error (mV)	62.8671	
Maximum difference (mV)	71.91 at 2.00 ms	
Time lag (ms)	1590	
**Kolmogorov–Smirnov Test**		
H-value	1 (distributions are different)	
*p*-value	<0.0001	
KS statistic	0.9844	

**Table 5 molecules-29-04700-t005:** Summary of phasic spiking characteristics in omeprazole–proteinoid complexes. This table presents a detailed comparison between the input signal from the Izhikevich neuron model configured for phasic spiking and the output from the omeprazole–proteinoid complex. The results reveal significant signal transformation by the omeprazole–proteinoid system. The input signal’s wide voltage range (−64.89 mV to 62.46 mV) is markedly compressed in the output (−2.25 mV to 3.17 mV), demonstrating strong attenuation. The shift from a negative input mean (−54.81 mV) to a positive output mean (0.54 mV) suggests a fundamental change in signal characteristics. The reduced standard deviation in the output (0.33 mV vs. 8.23 mV input) indicates a smoothing effect. The low correlation coefficient (0.4503) implies substantial non-linear processing, more pronounced than in other spiking modes. The negative time lag of −359 ms is particularly noteworthy, suggesting anticipatory behaviour in the omeprazole–proteinoid complex. This contrasts with the positive lag observed in induced spiking, highlighting mode-specific processing. The high RMSE (55.9366 mV) and maximum difference (67.00 mV) further quantify the extent of signal transformation. The Kolmogorov–Smirnov test results (KS statistic = 0.9945, *p* < 0.0001) indicate the most significant distributional difference among all spiking modes studied, emphasising the unique processing characteristics of the omeprazole–proteinoid system during phasic spiking stimulation.

Metric	Input Signal	Omeprazole–Proteinoid Output
Mean (mV)	−54.81	0.54
Standard deviation (mV)	8.23	0.33
Maximum (mV)	62.46	3.17
Minimum (mV)	−64.89	−2.25
**Comparative Metrics**		
Correlation coefficient	0.4503	
Root mean square error (mV)	55.9366	
Maximum difference (mV)	67.00 at 2.00 ms	
Time Lag (ms)	−359	
**Kolmogorov–Smirnov Test**		
H-value	1 (distributions are different)	
*p*-value	<0.0001	
KS statistic	0.9945	

**Table 6 molecules-29-04700-t006:** Summary of tonic spiking characteristics in omeprazole–proteinoid complexes. This table presents an analysis of the input signal from the Izhikevich neuron model configured for tonic spiking and the corresponding output from the omeprazole–proteinoid complex. The results reveal significant and unique signal processing by the omeprazole–proteinoid system under tonic stimulation. The input signal’s broad voltage range (−60.29 mV to 62.17 mV) is substantially compressed in the output (−2.41 mV to 3.63 mV), demonstrating potent signal attenuation. Notably, the mean potential shifts from a negative input (−37.47 mV) to a positive output (0.80 mV), the highest positive shift observed among all spiking modes, suggesting a robust baseline alteration in signal characteristics. The dramatic reduction in standard deviation (from 20.55 mV to 0.48 mV) indicates a strong smoothing effect, potentially filtering out the high-frequency components of the input signal. The correlation coefficient (0.6823) is higher than in phasic mode but comparable to induced mode, implying a more linear relationship between the input and output while still preserving significant non-linear processing. The negative time lag of −1231 ms is the largest among all modes, suggesting a highly pronounced anticipatory behaviour in the omeprazole–proteinoid complex under tonic stimulation. This could indicate the development of a strong predictive response mechanism during sustained, regular input. The root mean square error (43.2821 mV) is the lowest among all modes, suggesting that tonic spiking may induce the most consistent and predictable response in the complex. The Kolmogorov–Smirnov test results (KS statistic = 0.9276, *p* < 0.0001), while still indicating significantly different distributions, show the lowest KS statistic among all modes. This suggests that the output distribution in tonic mode, while distinct, may be closer to the input distribution compared to other spiking patterns.

Metric	Input Signal	Omeprazole–Proteinoid Output
Mean (mV)	−37.47	0.80
Standard deviation (mV)	20.55	0.48
Maximum (mV)	62.17	3.63
Minimum (mV)	−60.29	−2.41
**Comparative Metrics**		
Correlation coefficient	0.6823	
Root mean square error (mV)	43.2821	
Maximum difference (mV)	62.11 at 0.56 ms	
Time lag (ms)	−1231	
**Kolmogorov–Smirnov Test**		
H-value	1 (distributions are different)	
*p*-value	<0.0001	
KS statistic	0.9276	

## Data Availability

The data for the paper are available online and can be accessed at https://zenodo.org/records/13208375.

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
