# Peer review of "The Effects of Omeprazole on the Neuron-like Spiking of the Electrical Potential of Proteinoid Microspheres"

_molecules, 2024, doi:10.3390/molecules29194700_

Round 1

Reviewer 1 Report

Comments and Suggestions for Authors

The authors propose to characterize the behavior of neurons in the presence of a drug (omeprazole-proteinoid complex).  The main reason for doing this is to add the capability of neuromodulation using these drugs for the purpose of bio-synthetic computing. 

I was initially excited to read about this paper based on its subject, but as I tried to follow the logic, I became quite confused.

1.       Why use “Omperazole”, which is described as a PPI used to decrease production of acid in the gut – what does this have to do with neurons?

2.       Why use “Protenoids” in addition to this?

3.       Then I assumed that this study was going to be a computational model – that’s what the Izhikevich model is, but here it is used to drive an electrochemical experiment. So, as I read, based on Figure 3, the function generator “produces stimuli that resemble those experienced by neurons”.  I don’t know what “stimuli” mean in this case – is this membrane potential of a neuron?  Where is this voltage applied?  Figure 3 shows two electrodes – is this between the two electrodes?  Where is the recording made – same electrodes?  Why?  What do Pt and Ir electrodes and their interaction with the solution have to do with a model of a neuron with very specific membrane proteins that is affected by a chemical (“omeprazole-proteinoid”)?  Why is there no control to indicate what happens if some other chemical is used?      

Author Response

Reply attached

Reviewer 2 Report

Comments and Suggestions for Authors

1, line 61: Please explain the reason for comparing V with 30 mV, and also include in the text what happens if V is smaller than 30 mV.

2, Equation (1)-(2) require a reference.

3, table 1: please check the chemical formula of Omeprazole and Esomeprazole.  

4. Fig. 4-7 (a) and (b): What does the negative time indicate in Fig. 4 (a) and (b)? Also, is (b) a zoomed-in view of the red output signal in (a)? The output in (b) is so weak that it is difficult to distinguish from the noise. Please clarify whether this is a real signal or just noise.

5, Fig. 4-7c what is the unit of lag?

6, Fig.4-7 d Please label the red and blue curves to indicate what they correspond to.

7, Fig. 3: From Table 2, the mean output voltage is only 0.6 mV. Is an amplifier needed to read out this weak signal? If yes, it should be included in the Fig. 3.

8, line 129: The output signal is much weaker than the input signal, so comparing the deviations between the input and output is not reasonable.

9, line 132, 205, 348: Negative time seems unusual. It’s strange that the output occurs before the input.

10, Equations (1) - (3) and (11) - (13) are duplicated.

11, line 170: add the situation that v<30+ΔVop

Author Response

Reply attached

Reviewer 3 Report

Comments and Suggestions for Authors

The manuscript explore the impact of omeprazole-proteinoid complexes on Izhikevich neuron model. The work is interesting as it presents a potential material and device for neuromorphic computing.

The omeprazole is identified as a proton pump inhibitor, A brief explanation of why omeprazole was chosen and how it interacts with proteinoids may help readers understand more easily.

Typographical Error: The word "hybdrid" should be corrected to "hybrid."

Author Response

Reply attached

Round 2

Reviewer 2 Report

Comments and Suggestions for Authors

1, Units need to be added to the color bar in Fig. 9.

2, What do the two curves represent in Figs. 5-8 (d)? Please provide a brief explanation for why the blue curve drops or grows when the input quantities are above or below -70/70.

Reviewer 3 Report

Comments and Suggestions for Authors

Authors have addressed all my concerns,

Thank you,
